# Separation of soil respiration; a site-specific comparison of partition methods

Louis-Pierre Comeau[1,2], Derrick Y. F. Lai[1], Jane Jinglan Cui[1], Jenny Farmer[3]

[1] Department of Geography and Resource Management, Chinese University of Hong Kong, Shatin, Hong Kong.

[2] Fredericton Research and Development Centre, Agriculture and Agri-Food Canada, Fredericton, NB E3B 4Z7, Canada.

[3] Institute of Biological and Environmental Science, University of Aberdeen, 23 St Machar Drive, Aberdeen AB24 3UU, UK

*Correspondence to*: Derrick Y.F. Lai dyflai@cuhk.edu.hk

**Abstract.** Without accurate data on soil heterotrophic respiration (Rh), assessments of soil carbon (C) sequestration rate and C balance are challenging to produce. Accordingly, it is essential to determine the contribution of the different sources of the total soil $CO_2$ efflux (Rs) in different ecosystems, but to date, there are still many uncertainties and unknowns regarding the soil respiration partitioning procedures currently available. This study compared the suitability and relative accuracy of five different Rs partitioning methods in a subtropical forest: (1) regression between root biomass and $CO_2$ efflux; (2) lab incubations with minimally disturbed soil microcosm cores; (3) root exclusion bags with hand-sorted roots; (4) root exclusion bags with intact soil blocks; and (5) soil $\delta^{13}$C-$CO_2$ natural abundance. The relationship between Rh and soil moisture and temperature was also investigated. A qualitative evaluation table of the partition methods with five performance parameters was produced. The Rs was measured weekly from February 3[rd] to April 19[th] 2017 and found to average 6.1 ±0.3 Mg C ha$^{-1}$ y$^{-1}$. During this period, the Rh measured with the in-situ mesh bags with intact soil blocks and hand-sorted roots were estimated to contribute 49±7% and 79±3% of Rs respectively. The Rh percentage estimated with the root biomass regression, microcosm incubation and $\delta^{13}$C-$CO_2$ natural abundance were 54 ±41%, 8-17% and 61 ±39% respectively. Overall, no systematically superior or inferior Rs partition method was found. The paper discusses the strengths and weaknesses of each technique with the conclusion that combining two or more methods optimizes Rh assessment reliability.

# 1 Introduction

During the 2016 Convention of Parties (COP21) of the United Nations Framework Convention on Climate Change (UNFCCC) in Paris, the goal of increasing global soil organic carbon (SOC) stocks by 0.4 percent per year was set, with the aim of mitigating global anthropogenic greenhouse gas emissions (Minasny et al., 2017). This ambitious target was set based on the concept that the SOC in the top soil layer is sensitive and responsive to management changes and may offer opportunities to mitigate the current increases in atmospheric $CO_2$ concentration (McConkey et al., 2007). Of the carbon (C) that enters into ecosystems via photosynthesis, a fraction is directly respired by the roots and above ground plant parts (autotrophic respiration) to produce energy (i.e. adenosine-5'-triphosphate), with the other fraction synthesized into organic molecules. Some of these C-containing compounds are harvested or consumed by herbivores and the remainder is added to the soil as plant residues (Janzen et al., 1998). Subsequently, a portion of these fresh organic compounds are respired by organisms (heterotrophic respiration) and the other portion is converted into SOC by the genesis of soil organic matter (SOM) (Janzen, 2006; Lal, 2005). If the amount of new organic residues added to the soil is greater than the C lost by SOC decomposition, SOC content increases (Ellert and Bettany, 1995).

Typically, many years (up to decades) are needed to assess SOC stock changes over time in order to evaluate which management practices are beneficial for SOC sequestration (Harmon et al., 2011; Wood et al., 2012). This timeframe is impractical for policy makers to evaluate the mitigation potential of different land management practices, in particular with the pressing need of the UNFCCC goal of increasing the global SOC stocks by 0.4 percent per year. An alternative approach that allows a more rapid evaluation of these long term impacts is to combine the SOC stock change procedure (e.g. VandenBygaart et al., 2008) with the soil C efflux balance approach (i.e. Hergoualc'h and Verchot, 2011), which although demanding and with some uncertainties can provide results on soil dynamics over an annual basis. The soil C efflux balance approach involves calculating the rate of C entry and exit in the soil. However, the total $CO_2$ efflux (Rs) from soil does not provide the necessary information to estimate whether the soil is a net source or net sink for atmospheric $CO_2$ (Kuzyakov and Larionova, 2005). Total soil efflux is a combination of root based respiration (autotrophic (Ra)) and heterotrophic respiration (Rh). Autotrophic respiration does not contribute to net C losses to the atmosphere as it is cycled within the ecosystem, whereas Rh represent net C losses. However, the boundary between Ra and Rh is not easy to distinguish (i.e. the rhizo-microbial respiration is linked to both) and realistic Rh assessments are difficult to produce (Braig and Tupek, 2010).

Reviews of Rh-Rs segregation methods have been made (e.g. Kuzyakov, 2006) but no site specific study has been made analysing several different partition techniques simultaneously. The goal of our study was to compare partitioning methods to separate $CO_2$ efflux into its Rs and Rh component in a subtropical secondary forest in Hong Kong. Five methods were selected based on their suitability in the studied ecosystem. Three methods were traditional techniques (i.e. regression between root biomass and $CO_2$ efflux; root exclusion bags with hand-sorted roots; and soil $\delta^{13}C$-$CO_2$ natural abundance) and two were innovative variation of existing methods (i.e. root exclusion bags with intact soil blocks; and lab incubations with minimally disturbed soil microcosm cores). The influence of soil moisture and temperature on $CO_2$ efflux was also analyzed.

## 2 Methodology

The research was conducted in a subtropical secondary forest of Hong Kong (Tai Po Kau Nature Reserve; 22° 27′N, 114° 11′E). The landscape is typical of the escarpment of the Tai Mo Shan mountain range, the system formed by volcanic activities in the Late Jurassic epoch (Langford et al., 1989). The rocks are mainly rhyodacite to rhyolite from the Tsuen Wan Volcanic Group (Davis et al., 1997). The study site was approximatively 600 m. above sea level and the slope surfaces were stable and vegetated. The forest was approximatively 50 years old and was covered with continuous canopy. More than 100 plant species were registered in the Nature Reserve. The following genera were found in the study area: *Machilus sp, Meliosma sp, Garcinia sp, Engelhardia sp, Psychotria sp, Ilex sp, Eurya sp and Lithocarpus sp*. The mean annual temperature was 23.3°C and annual precipitation 2400 mm with a hot-humid season (April–September) and a cool-dry season (October–March) (Hong Kong Observatory). The study area was 1 ha and was located inside a long-term research site belonging to the Chinese University of Hong Kong. The canopy was closed in the area with an average solar radiation at 2 m high of 13.8 W/m² (non-published data).

### 2.1 Soil general characterization

Four soil profiles were dug in the study area, characterizing the different landforms (i.e. back-slope and foot-slope) present at the site. Morphological description was done according to Jahn (2006) and the soil was classified with the World Reference Base (IUSS-Working-Group-WRB, 2014). Soil pH was determined with a glass–calomel electrode pH meter (McLean, 1982). Rainfall and air temperature were recorded hourly with a HOBO Weather station (rain gauge, S-RGB-M002; air temperature/RH, sensor S-THB-M008, Onset Computer Corp., USA). Water holding capacity was assessed by saturating the soils, allowing them to freely drain for 24 h and determining gravimetric water

content after oven-drying at 105 °C following Arcand et al. (2016). Root biomass was measured by collecting soil cores (inner diameter 5 cm, height 5 cm) and determined using the approach of Tufekcioglu et al. (1999). The soil was dried, finely ground, and subsequently analyzed for total C and N content using a CNS Analyzer System (Perkin Elmer 2400 Series II CHNS/O Analyzer, USA) at the Earth System Science Laboratory of the Chinese University of Hong Kong (ESSL-CUHK).

**2.2 Partitioning soil respiration**

To produce estimates of %Rh, five different approaches were designated based on their suitability in the study area. We decided to use three customary methods often employed in this type of ecosystem (i.e. regression between root biomass and $CO_2$ efflux; root exclusion bags with hand-sorted roots; and soil $\delta^{13}$C-$CO_2$ natural abundance) and two innovative variation of existing techniques (i.e. root exclusion bags with intact soil blocks; and lab incubations with minimally

disturbed soil microcosm cores).

2.2.1 Regression between root biomass and $CO_2$ efflux method

The root biomass regression technique is based on the relationship between the $CO_2$ emitted by the root-rhizosphere and root biomass and the $CO_2$ efflux slope fitted from SOM decomposition (i.e. Rh), corresponding to the intercept of the linear regression line (Kucera and Kirkham, 1971). This method was made following Farmer (2013) with 22

sampling spots. Each spot was a square of 20 x 20 cm randomly distributed in the study area. In each spot, Rs was determined per triplicate using a portable IRGA as described above. Concurrently with $CO_2$ efflux measurements, air and soil (10 cm depth) temperatures and soil volumetric moisture content were measured at each sampling spot. Immediately after the Rs measurement, the 20 x 20 cm squares were excavated to 25 cm depth. All the visible roots (diameter larger than 0.1 cm) from the excavated soil were collected. In the lab, the roots were washed and then oven

dried at 60°C until a steady dry weight was attained, which was then recorded. A linear regression analysis between root quantity and $CO_2$ efflux was performed using the program R Foundation for Statistical Computing version 2.8.1 (R Development Core Team, 2008).

2.2.2 Lab incubations

For the lab incubations, undisturbed soil cores with a volume of 98 cm$^3$ (inner diameter 5 cm, height 5 cm) were

105 collected using a stainless-steel core soil sampler from the upper part of the soil profile (0–5 cm). In the study area, four groups of four soil cores (stratified random design) were collected, then pooled per group and brought to the lab (Fig.

1). Subsequently all visible roots were removed but with special care to not destroy the micro-aggregates. The soil was then repacked to original bulk density in minimally disturbed soil microcosm cores of 45 cm$^3$ (inner diameter 3.5 cm, height 5 cm). The soil cores were separated in four groups of different volumetric moisture content (i.e. 15, 25, 35 and 45). These moisture levels corresponded to the natural annual fluctuation in the field (i.e. from dry to moist season) (Cui and Lai, 2016). After moisturizing the samples with distilled water, each individual soil core was placed into a hermetically sealed 2.9 dm$^3$ plastic container and left to stabilize in dark for two weeks at 25°C. After that, the experiment lasted four weeks and had four different incubation temperature levels (one per week; 14°C, 20°C, 26°C and 32°C) corresponding to the minimum, intermediate and maximum soil temperature values in the field based on preliminary studies (Cui and Lai, 2016). At the beginning of each week, the soil cores were pre-incubated in their incubation box to their corresponding weekly temperature (i.e week #1, 14°C … week #4, 32°C) for 3 days and then opened and vented for one minute. From all the boxes gas samples were collected (20 ml) with an air-tight syringe (t= 0, 24, 72 hour) after box closure. The $CO_2$ concentrations were analyzed within 48 hours with a gas chromatograph (GC system 7890A, Agilent Technologies). The GC system was equipped with a flame ionization detector and an electron capture detector to quantify $CO_2$ concentration. Between each measurement session, the boxes were opened to vent and the moisture of the soil cores was re-adjusted if needed.

Gaussian 3D regression fitted curve was derived as shown in equation 1. using SigmaPlot version 10.0 (Systat Software, San Jose, CA).

$$f(x, y) = a \times \exp\left[-0.5 \times \left(\left(\frac{x-xo}{b}\right)^2 + \left(\frac{y-yo}{c}\right)^2\right)\right] \tag{1}$$

where $f(x,y)$ is the $CO_2$ efflux function; $a$, $b$ and $c$ are constant coefficients; x is the soil temperature (°C); y is the soil moisture content (%); $x_0$ is the average temperature; $y_0$ is the average soil moisture.

2.2.3 Root exclusion bag methods

To partition the $CO_2$ efflux in-situ into Rs and Rh using mesh bags, two different approaches were followed: 1) the traditional dug soil with hand-sorted root removal and refilling method (HS) (Fenn et al., 2010; Hinko-Najera, 2015) and 2) a variant of it with intact soil blocks (IB). The HS method consisted of digging a pit for each bag with a size matching the bag dimensions (20 × 20 cm, depth: 25 cm) where the soil was excavated in layers (to maintain soil horizons) and visible roots were removed before repacking the bag inside the pit with the removed soil. The IB variant

of this technique consisted of extracting a cube as intact as possible from the soil ($20 \times 20$ cm, depth: 25 cm). This was then, tightly placed into the micromesh bag and inserted back into its original pit. For both methods, the same type of micromesh bags (38μm nylon mesh), closed at the bottom but open at the top were used. This mesh size was used to impede roots from entering inside the bags, but allowed mycorrhiza to penetrate (Moyano et al., 2007). In all the pits excavated, no root below 25 cm depth was observed. Collars measuring 10 cm diameter were installed on the soil in the center of each bag to a depth of 8 cm, for heterotrophic emissions sampling.

Seven plots were randomly distributed inside the study area. In each plot, an IB bag was paired with a HS bag with space of 150 cm between them. The root exclusion bags were installed during the month of October 2016 and were let to stabilize for three months. At 1 m distance from each root exclusion bag, a collar was inserted into non-disturbed soil to measure Rs. To assess Rs and Rh without the influence of litterfall decomposition, the collars were cleared of leaves and flowers on a weekly basis.

From February 3$^{rd}$ to April 19$^{th}$ 2017 the collars were measured weekly with an IRGA (Environmental Gas Monitor, EGM-4, PP Systems, UK) attached to a soil respiration chamber (SRC-1, PP Systems, UK). Soil temperature and soil moisture were measured in the area located between the collar and the edge of the bag (to 10cm depth, HH2, Delta-T Devices, Cambridge-England). At the end of the study all the root exclusion bags were removed from the soil and inspected to ensure that no root had penetrated inside. The soil inside the measurement collars was then collected to assess bulk density (van Reeuwijk, 1992). Mathematical calculation and descriptive statistical analyses were done with Microsoft Excel XP$^{®}$.

### 2.2.4 $\delta^{13}$C natural abundance method

Millard et al. (2010) have demonstrated that the natural abundance $\delta^{13}$C (‰) of Rs falls between the $\delta^{13}$C values of the Rh and Ra. The $\delta^{13}$C of Rs and Rh respiration was determined following Lin et al. (1999) and Millard et al. (2010). The isotopic partitioning experiment assessed values of the $\delta^{13}$C of the Rs, Ra and Rh. The sampling took place on March 15$^{th}$ 2017. A closed chamber (10 cm diameter, 10 cm high) was positioned on each emissions measurement collar (n=7 as described in section 2.2.3). The chambers were flushed for 2 minutes with $CO_2$-free air to remove all the atmospheric air trapped within the headspace. Chambers were left to incubate for 40 minutes to ensure the concentration of the chamber sample reached above 400 ppm of $CO_2$ from which a duplicate sample of the gas in the chamber headspace were extracted into evacuated vials to give the $\delta^{13}$C of the Rs. Subsequently, the soil under the

160 chamber was dug and immediately brought to the lab (ESSL-CUHK, less than 30 minutes travel) where the soil and the roots were carefully separated. The roots were gently washed with water to remove adhered soil aggregates and slightly dried afterward with paper towels. Samples of 5 g of root and 10 g of root-free soil per chamber were incubated in $CO_2$ free air in 250 ml airtight glass bottles to give the $\delta^{13}C$ of the Ra and Rh respectively. The bottles were left to incubate for 90 minutes before duplicate extraction into evacuated vials. As recommended by Midwood et al., (2006), before gas

sample extraction, the butyl rubber septa used to seal the vials were heated at 105°C for 12 h. The C isotope ratio of the $CO_2$ in all samples was analyzed using a Gas-bench II connected to a DeltaPlus Advantage isotope ratio mass spectrometer (both Thermo Finnigan, Bremen, Germany) at the James Hutton Institute Scotland UK. The $\delta^{13}C$ ratios, all expressed relative to Vienna-Pee-Dee Belemnite (VPDB), was calculated with respect to $CO_2$ reference gases injected with every sample and traceable to International Atomic Energy Agency reference material NBS 19 TS-

Limestone. Measurement of the individual signatures of the natural abundance $\delta^{13}C$ of the Rs, Rh and Ra allowed partitioning between the different sources using the mass balance mixing model (Lin et al., 1999; Millard et al., 2010):

$$\%Rh = 1 - \frac{\delta Rs - \delta Rh}{\delta Ra - \delta Rh} \times 100 \qquad (2)$$

where *%Rh* is the proportion of Rh from Rs, and *δRs*, *δRh* and *δRa* are the $\delta^{13}C$ isotopic signatures.

### 2.3 Qualitative comparison of segregation methods

While there is much debate in the literature, and methods are still being developed, isotopic partitioning methods are increasingly being recognized as a more accurate approach to segregation of Rs than non-isotopic techniques (Paterson et al., 2009; Kuzyakov, 2006). Thus, for comparison purposes we used the soil $\delta^{13}C$ natural abundance method as a reference point for segregation relative accuracy. Partition methods that had Rh%: <10, 10-20 and >20 lower or larger than the $\delta^{13}C$-$CO_2$ natural abundance were categorized as high, intermediate and low relative accuracy,

respectively. The level of precision of the segregation methods was determined with the statistical variance associated with the Rh/Rs ratio averages. High, intermediate and low precision were attributed to Rh% standard errors of <10, 10-20 and >20, respectively. The level of complexity was evaluated with the number of steps required to complete each method. For example, the hand-sorted root exclusion bags technique was judged as a four steps method (pit excavation, root removal, bag/pit refiling, and $CO_2$ efflux measurements). Methods with five steps or less were deemed simple and

six steps or more deemed as complex. The time needed to set up the experiment was assessed by counting the number

of working hours (eight hours equal one day) required prior to the start of the measurements. The time needed to produce seasonal trends was the number of months of measurements (in the field or in the lab) required to characterize the Rh at the different temperature and moisture levels of the year.

## 3 Results

### 3.1 Soil characteristics

According to their morphology and diagnostic properties, the soil was classified as Alic Umbrisol (Nechic) and Haplic Alisol (Nechic) (IUSS-Working-Group-WRB, 2014). The difference between the two soil groups was the thickness of humus-containing horizon (between 20 and 30 cm for the Umbrisol; while, 10 to 20 cm for the Alisol). The A horizon had high organic C content (3.2 ±0.2%) and high acidity (pH $_{H2O}$ 4.2) (Table 1). The sub-surface soil was represented by clayey yellow-colored profiles with an argic horizon. Soil texture was heavier in the argic horizon than in the topsoil and parent material. The structure in all the soil profiles was predominantly granular in the upper horizons, whereas the argic horizon was characterized by subangular blocky structure (Table 1). The argic horizon was deemed to be of high-activity clays and low cation base status based on previous results in the area, along with soil acidity, type of parent material and level of mineralization of the bedrock in the soil pits.

### 3.2 Regression between root biomass and $CO_2$ efflux

The 22 quadrats used for the root biomass regression assessment yielded an average Rs of 11.1 ±1.0 Mg $CO_2$–C ha$^1$ y$^{-1}$. The regression of the $CO_2$ efflux against root biomass produced a statistically significant slope correlation of 0.08 ±0.04 g $CO_2$ m$^2$ h$^{-1}$ per mg root cm$^{-3}$ (p=0.03), and set the intercept at 0.25 ±0.10 g $CO_2$ m$^2$ h$^{-1}$ (p=0.02) which represented the basal efflux in the absence of roots i.e. the Rh (Fig. 3). The Rh measured (i.e. slope intercept) when the root biomass regression technique was performed (October 2016) was 6.0 ±2.4 Mg C ha$^{-1}$ y$^{-1}$, equivalent to 54% of the Rs (Table 5).

### 3.3 Lab incubation

During the incubation with minimally disturbed soil microcosms, the average (moisture levels combined) $CO_2$ efflux at 14, 20, 26 and 32°C was 0.36 ±0.50, 0.67 ±0.38, 1.40 ±0.91 and 2.24 ±1.39 Mg $CO_2$–C ha$^1$ y$^{-1}$, respectively (Fig. 4). The exponential relationship between $CO_2$ efflux, soil temperature and moisture is presented in Table 3.

**3.4 Root exclusion bag methods**

During the root exclusion bags measurements period (Feb-Apr 2017), the average air temperature was 16ºC and the total rainfall 107 mm. During that period the Rs averaged 6.1 Mg C ha$^{-1}$ y$^{-1}$ (Fig. 2). One of the requirements for the suitability of root exclusion bag methods to estimate Rh is that soil bulk density, soil temperature and moisture are statistically equal inside and outside of the bags. In this experiment, no significant differences were detected regarding the bulk density and soil temperature (p=0.87 and p=0.15, respectively) but the volumetric soil moisture in the HS bags was on average 17% lower than outside the root exclusion bags (p=0.04) (Table 2). As would be expected, all Rh IB and Rh HS efflux rates were lower than the Rs efflux at each measurement date. Throughout the experiment, the Rh IB was consistently lower than the Rh HS except on March 31$^{st}$ (Fig. 2b).

**3.5 Soil $\delta^{13}$C-CO$_2$ natural abundance**

On a land-scape basis, the $\delta^{13}$C-CO$_2$ natural abundance method reasonably segregated the three respiration components (Table 4). The $\delta^{13}$C-CO$_2$ of the Rh HS, Rh IB, and Rh lab were in a very close range (i.e. -16.52 to -16.75), indicating that the efflux measured in the root exclusion bags were not contaminated with root respiration. Based on the $\delta^{13}$C-CO$_2$ of the Rs (-18.21±0.53), the Rh lab (-16.75±0.54), and the Ra lab (-20.44±0.65) the average percentage of heterotrophic respiration was 61 ±39% (Table 5).

Using the $\delta^{13}$C-CO$_2$ method as a base-line, the increase/decrease of the Rh from root biomass regression, lab incubation, hand-sorted and intact block (IB) root exclusion techniques were -11%, -72-87%, +30% and -20% , respectively (Table 5).

**4 Discussion**

**4.1 Regression between root biomass and CO$_2$ efflux technique**

As demonstrated by Gupta and Singh (1981) the intercept of the regression line between the independent variable (i.e. root biomass) and the dependent variable (i.e. Rs) corresponds to soil respiration in absence of root (i.e. Rh) (Fig. 3). In this study the regression had ten points (45%) outside the confidence interval but the intercept and slope were still statistically significant. This uncertainty in the regression fit was likely caused in large part by the older roots which are bulkier but respire less than fine and young roots (Behera et al., 1990). However, this method had the closest average Rh/Rs ratio to the $\delta^{13}$C natural abundance technique. Consequently the root biomass regression technique was assessed as high relative accuracy and low precision (Table 6). Previous studies also highlighted large variation of CO$_2$

efflux and root biomass which causes relatively low coefficient of determinations (Behera et al., 1990; Farmer, 2013). In accordance to Kuzyakov (2005), this method was comparatively simple (Table 6).

**4.2 Lab incubation method**

Interpreting soil respiration processes in response to seasonal changes is generally challenging because soil temperature and moisture regularly covary (Carbone et al., 2011; Davidson et al., 1998). The lab incubation technique was the only method capable of dividing the effect of soil temperature and moisture on Rh and to produce a significant Gaussian regression fit (Table 3). However, the microcosm incubation produced Rh values notably lower than the other techniques (Table 5). This may be due to three different causes. First, the fact that the soil column in the incubation microcosms were 5 cm thick while the A horizon in the field (i.e. where the Rh assessments from the other techniques were made) was 10 cm (Table 1). Second, to prevent potential shifts in the microbial community during the incubations (i.e. adapting to lower resource availability), prior to the beginning of the experiment the microcosms were left to stabilize for two weeks. Accordingly, the fresh and labile organic residues that would in the other segregation methods contribute to the soil respiration had already decomposed before the beginning of the incubations. Third, the low Rh of the lab incubation method could also be attributed in part to the fact that this technique did not contain any rhizomicrobial respiration and its priming effect (Kuzyakov et al., 2000). That is, this method produced Rh from basal microbial respiration which is considered to be from stabilized SOM with slow turnover rates (Kuzyakov, 2006 Neff et al., 2002). In view of that, with additional field and lab methods development it would be possible to further segregate Rh assessments into percentage of rhizomicrobial respiration, decomposition of plant residues and basal decomposition of SOM. Overall, the lab incubation technique was slightly more complex than the non-isotopic field Rh assessment methods but allowed a prompt determination of Rh whilst simulating year-round field conditions (Table 6). Further studies should test the effect of microcosm height on Rh in relation to field measurements and determine microbial C use efficiency by isothermal microcalorimetry during the incubations.

**4.3 Root exclusion bags methods**

The HS and IB methods had %Rh of 79 ±3 and 49 ±7 %, respectively. The variances around the means were the lowest of all the field segregation methods tested (Table 5). Comparing the %Rh of the HS and IB with the $\delta^{13}C$ natural abundance technique, they resulted 18% above and 12% below, respectively. Thus the root exclusion bags methods

were judged of intermediate relative accuracy and high precision. Also, the HS and IB methods were fast and simple to setup (Table 6).

The micromesh size used in the root exclusion bags was 38μm which was reported to impede root penetration but to allow arbuscular mycorrhizal to spread inside the bags (Moyano et al., 2007; Rühr and Buchmann, 2010). In turn, Fenn et al. (2010) stated that in the mycorrhizal structures the arbuscules exist within roots, and therefore, the $CO_2$ efflux from these bags could contain some portions of the roots respiration. Contrary to this, the IB and HS air samples analyzed for $\delta^{13}C$ had an isotopic signature close and not statistically different from the gas samples collected in the lab airtight glass bottle of fresh soil without roots. This indicates that the root exclusion bags (both IB and HS) did not comprise traces of root respiration that had a significantly larger $\delta^{13}C$-$CO_2$ signature (Table 4). After the three month period of soil stabilization, both bag methods for partitioning total soil respiration and root-free soil respiration components successfully produced Rs>Rh in every sampling dates indicating that efflux rates within the bags had reached an apparent post disturbance state (Fig. 2). Also, in both IB and HS, soil temperature and bulk density were statistically equal to the surrounding soil (i.e. Rs) (Table 2). This indicates that the environmental conditions inside and outside of the bags were similar in respect to these two parameters. However, the soil moisture of the IB was statistically equal to the surrounding soil but for HS it was significantly lower. This was likely caused by the breakdown of the original soil structure at the moment of root removal that increased the drainage inside the HS bags. Moyano et al. (2007) also found that soil moisture can be affected by the hand-sorted root exclusion bag method. Overall, HS had a moisture level 20% lower and an Rh efflux 60% larger than IB (Table 5). The larger HS Rh efflux compared with IB Rh could be due in part to the lower soil moisture in the former. This likelihood is supported by the fact that in the regression fit the maximum Rh was when moisture content was relatively low (i.e. y0, Table 3). In addition, the breakdown of numerous soil aggregates during the root removal likely allowed the soil microorganisms to thrive in previously encrusted SOM domains of the HS soil. It has been shown that the part of the SOM that is located in the interior of the soil aggregates is hardly accessible to microorganisms, and thus not easily mineralized unless the aggregates are shattered (Goebel et al., 2005).

### 4.4 Soil $\delta^{13}C$ natural abundance method

The three respiration components of this method (i.e. $\delta^{13}C$-$CO_2$ from Rs, Rh and Ra) had large standard errors (Table 4) that produced a high uncertainty value in the Rh/Rs ratio assessment (61 ±39 %, Table 5). This method was accordingly deemed of low precision (Table 6). This, in turn, impeded our ability to produce an Rh/Rs ratio assessment

in the individual collars. The low precision of this method indicates that some biases in the assessment of relative accuracy could potentially have been generated. This large $\delta^{13}$C-$CO_2$ variance was likely caused by variability of $\delta^{13}$C in soil and plants residues and also due to $^{13}$C discrimination by plants that is affected by moisture content and nitrogen availability (Hogh-Jensen and Schjoerring, 1997). In addition, other studies reported the variability of $\delta^{13}$C in soil or plants of at least 1–2‰, which in some cases can limit the capacity to produce soil respiration segregation assessments (Accoe et al., 2002; Cheng, 1996; Farquhar et al., 1989). Because soils are porous mediums, excluding any atmospheric $CO_2$ that has a different isotopic composition (i.e. $\delta^{13}$C -7.5 to -8.5 ‰) to that of the Rs efflux is challenging and potential air contaminations have to be considered when analyzing the results (Millard et al., 2010). In our study, the Rh $\delta^{13}$C was measured in the field (IB and HS; potentially air contaminated) and from airtight containers in lab incubations of root free soil (Rs lab; not potentially air contaminated). Both ways produced $\delta^{13}$C in a close range and without statistical differences between them (Table 4). This indicates that the chamber system used in the field to collect the $\delta^{13}$C efflux samples was adequately effective to prevent air contamination. Overall, the soil $\delta^{13}$C natural abundance method was fast to setup but was relatively complex to perform with a field and lab component to be accomplished within a short period of time (Table 6). Further studies should use a large amount of sampling points to attempt reducing the respiration components standard errors.

### 4.5 Comparison of methods and recommendations

The analysis of the five different Rs partitioning methods examined in this study shows that none of them was fully satisfactory. That is, each technique had strengths and weaknesses (Table 6). Using $\delta^{13}$C-$CO_2$ is acknowledged as the preeminent way to segregate Rs (Cheng, 1996; Kuzyakov, 2006); and accordingly the relative accuracy of the other methods was defined by their difference with this method. However, we recognize this was a precarious approach because the $\delta^{13}$C-$CO_2$ method had a large variation. The root biomass regression, which is also recognized in the literature as a reliable method (Kuzyakov, 2006), gave a similar %Rh estimate. However, we found several other shortcomings with the $\delta^{13}$C-$CO_2$ method. First, the conjunction of field and lab procedures makes it difficult to complete this method in one day as needed. Second, the air flushing with $CO_2$ free gas in the field (to prevent ambient $\delta^{13}CO_2$ contamination) makes that technique more complex than the other methods to assess Rh%. Third, the ability to perform this technique in remote areas is limited because the $\delta^{13}$C-$CO_2$ needs to be quickly assessed with a calibrated and accurate spectrometer (Midwood et al., 2006). Fourth, the large variation in $\delta^{13}$C-$CO_2$ of the respiration components (i.e. Ra, Rh and Rs) impeded the assessment of Rh% per individual collar. Accordingly, further studies

should analyze the spatial relationships of $\delta^{13}$C-CO$_2$ with soil properties and root characteristics. As standalone, the

$\delta^{13}$C-CO$_2$ technique was unable to produce an assessment of soil CO$_2$ efflux; and thus needed to be performed in

conjunction with field Rs measurements. In this regard, the $\delta^{13}$C-CO$_2$ complemented well with the root exclusion bags

methods because it allowed us to determine if the buried bags had teared and been invaded by roots and to standardize

Rh% determination.

The root biomass regression method had the advantage of being simple, and produced an average Rh% close to the

$\delta^{13}$C-CO$_2$ natural abundance. However it had the disadvantage of requiring a high number of replicates due to low

coefficient of determination between CO$_2$ efflux and root biomass. Another disadvantage of the root biomass

regression technique is that in order to produce seasonal trends, the labor intensive procedures (i.e. pit digging, CO$_2$

measurements and root counting) need to be reinitiated several times during the year. This shortcoming can be

particularly impractical in small plot experiments. Complementary studies should assess thresholds of root size to be

included in the regression fit in order to optimize the correlation fit and use the $\delta^{13}$C-CO$_2$ natural abundance method to

determine the effect of root size on the isotopic signature.

The root exclusion bags methods (i.e. HS and IB) had the advantage of being easy to monitor throughout the year,

capturing temporal variability of the % of Rh. The bags methods can be considered as a miniaturization of the

traditional soil trenching method. However, contrasting with large trenches (e.g. Comeau et al., 2016; Fisher and Gosz,

1986) the root exclusion bags had the advantage of being simpler to establish and allowed mycorrhiza development

inside the mesh bags (Moyano et al., 2007). Conversely, due to the relatively small bag sizes, root webs on the outside

edge could potentially contaminate Rh assessment. In this study, the $\delta^{13}$C-CO$_2$ determination made with the collars

located in the center of the bags showed no isotopic signature of root respiration. Similar to a trenching method, the

root exclusion bag method had the disadvantages of requiring several months of soil stabilization before CO$_2$ efflux

measurements could begin. Compared with the $\delta^{13}$C-CO$_2$ natural abundance method, the HS and IB overestimated and

underestimated %Rh by 18 and 12%, respectively. The divergences were likely caused by soil disturbances, alteration

in root demise dynamic and lack of root exudates. Correspondingly, Carbone et al. (2016) found 11% difference in

Rh% assessment between an isotopic partition method and the trenching technique. Comparing the HS and IB, the

former created more soil disturbances but the latter would not be suitable for soil with a high amount of sand, gravel or

rock because the intact blocks would collapse.

The lab incubations of the minimally disturbed microcosms was the only method that had absolutely no influence of root or mycorrhiza. Thus the results from this method exclusively represented the $CO_2$ efflux originating from the mineralization of the slow turnover SOC pool (i.e. basal soil respiration) (Pell et al., 2006). Assessment of basal soil respiration in relationship with the total Rh is of great importance in evaluating the dynamic of the stabilized SOC. In this study, the Rh% from the lab incubation was 8-17% while the $\delta^{13}C$-$CO_2$ natural abundance had an average of 61% Rh. Thus, if the soil incubation results were not affected by the height of the soil columns (as discussed above), basal respiration represented approximately one fifth of the Rh. Because stabilized SOC is a key indicator of soil quality and health (Creamer et al., 2014), further research should study the relationship between basal soil respiration and rhizosphere derived Rh. Also, future studies on soil $CO_2$ efflux segregation should include other partitioning techniques like the trenching method and the radiocarbon-based assessment (Chiti et al., 2015). Overall, results from field experiments exhibited a range of potential Rh between 2.5 and 6.0 Mg $CO_2$-C ha$^{-1}$ y$^{-1}$. With the publication of the total annual live biomass growth (i.e. including root and above-grown biomass) at the study site (Tai Po Kau Nature Reserve) assessment of net ecosystem C balance would then be possible.

**5 Conclusions**

Methods for determining ecosystem C fluxes need to be improved and applied to allow a quantitative understanding of the biological processes underlying SOC balance. This study compared five methods to assess Rh and our results showed large variation in effluxes and Rh/Rs ratio between the different techniques analyzed. The data revealed that the hand-sorted root exclusion bags and the intact root exclusion bags methods produced similar Rh efflux values and these efflux were slightly lower than the one produced by the root biomass regression method but notably larger than the lab incubation with soil cores. We found that methods with higher relative accuracy (soil $\delta^{13}C$-$CO_2$ natural abundance and root biomass regression) had lower precision (i.e. large variance) and methods with higher precision (root exclusions bags and lab incubation) had lower relative accuracy. Based upon these results, we suggest that when assessing rate of heterotrophic emissions and their contribution to total soil based emissions, two or more methods should be performed to produce more integral assessments.

**Acknowledgements**

The work described in this paper was supported by two grants from the Research Grants Council of the Hong Kong Special Administrative Region, China (CUHK14302014 and CUHK14305515), as well as a Direct Grant of CUHK

(SS16914). Thomas Lui is thanked for assistance with the gas chromatograph analysis. Jack Lin is thanked for the
assistance in building the microcosm tubes and for ensuring that the incubation containers were fully airtight.
Amelia Amadeo, Christoph Hartmann and Merve Oztoprak are thanked for the assistance during the field
measurements. Dr. Barry Thornton from the James Hutton Institute Scotland UK is thanked for the $\delta^{13}$C-$CO_2$
determination.

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

Table 1: Morphological description of the soil profiles at the study site

| Horizon depth (cm) | Color (dry) | Color (moist) | Field texture[a] | Structure[b] | Rock fragments volume % | Roots mg root cm$^{-3}$ | pH (H$_2$O) | pH (KCl) | WHC (g H$_2$O g soil$^{-1}$) | Soil organic C; N (%) |
|---|---|---|---|---|---|---|---|---|---|---|
| A 0~10 | 10 YR 6/2 | 10 YR 3/2 | SL | gr | 0 | 9.3 ±3.4 | 4.2 | 3.3 | 0.50 ±0.01 | 3.2 ±0.2; 0.24 ±0.02 |
| AB 10~25 | 10 YR 6/3 | 10 YR 4/2 | SL | gr | 0 | 1.6 ±0.6 | 4.5 | 3.9 | - | - |
| Bt 25~60 | 10 YR 8/8 | 7.5 YR 6/8 | CL | sbk | 10-20 | 0.5 ±0.2 | 4.7 | 4.0 | - | - |
| C 60~100 | 10 YR 8/8 | 10 YR 7/8 | CL | sbk | >60 | 0 | 4.7 | 4.0 | - | - |

[a] SL, sandy loam; CL, clay loam.
[b] gr, granular; sbk, subangular blocky.
WHC, water holding capacity
C, carbon; N, nitrogen

Table 2: Comparison of environmental parameters inside and outside the root exclusion bags

| Method | Soil temperature (°C) | Soil moisture (vol. %) | Bulk density (g cm$^{-3}$) |
|---|---|---|---|
| Inside hand-sorted root exclusion bags (HS) | 22.4 (0.2) α | 20.5 (1.2) β | 1.16 (0.04) α |
| Inside intact root exclusion bags (IB) | 22.6 (0.3) α | 25.5 (1.4) α | 1.13 (0.05) α |
| Outside root exclusion bags (Rs) | 22.4 (0.2) α | 24.8 (0.8) α | 1.14 (0.03) α |

Values are means and standard error. Values in the same column followed by a different Greek letter (α, β) are significantly
different from each other at α=0.05.

Table 3: Parameter values of the Gaussian 3D regression fitted curve (equation 1)

| Efflux | Parameter a (Mg ha$^{-1}$ y$^{-1}$) | Parameter xo | Parameter yo | Parameter b | Parameter c |
|---|---|---|---|---|---|
| Rh lab incubation | 5.0 *** | 49.2 *** | 34.7 *** | 15.7 *** | 19.2 *** |
| Rs field | 10.3 ** | 24.9 ** | 18.3 NS | 9.6 NS | 15.8 NS |
| Rh IB | 5.7 NS | 21.93 NS | 21.4 NS | 4.8 NS | 13.4 NS |
| Rh HS | 8.1 NS | 21.7 NS | 9.5 NS | 5.1 NS | 14.2 NS |

Rh lab incubation, heterotrophic respiration from the soil cores incubation
Rs field, total soil respiration from outside of the root exclusion bags
Rh IB, heterotrophic respiration from the intact root exclusion bags
Rh HS, heterotrophic respiration from the hand-sorted root exclusion bags
** and *** significant at $p < 0.01$ and $p < 0.05$, respectively; NS non-significant.
Parameters from equation 1. Parameter "a" correspond to the height of the maximum high of the curve (g $CO_2$ m$^{-2}$ h$^{-1}$); "xo" is
the peak soil temperature point (°C) in the curve, "yo" is the peak soil moisture (%) point in the curve, and "b" and "c" are the
Gaussian root mean squared widths of the soil temperature and soil moisture, respectively.

Table 4: Average $\delta^{13}$C-CO$_2$ results

| Method | $\delta^{13}$C-CO$_2$ (‰) |
|---|---|
| Rs[a] | -18.21 (0.53) αβ |
| Rh HS[b] | -16.65 (0.44) β |
| Rh IB[c] | -16.52 (1.07) β |
| Rh lab[d] | -16.75 (0.54) β |
| Ra lab[e] | -20.44 (0.65) α |

[a] Rs, gas samples collected from the field total soil respiration collars.
[b] Rh HS, gas samples collected from the field hand-sorted root exclusion bags collars.
[c] Rh IB gas samples collected from the field intact blocks root exclusion bags collars.
[d] Rh lab, gas samples collected from lab incubations of soil with freshly removed roots.
[e] Ra lab, gas samples collected from lab incubations of the roots extracted in Rh lab.
Values are means and standard error, n = 14 for Rs and Ra and n = 7 for HS, IB and Rh lab.
Values followed by a different Greek letter (α, β) are significantly different from each other at α=0.05.

Table 5: Comparison of heterotrophic respiration assessment methods

| Method | Rh efflux[a] | Rs efflux[b] | Rh / Rs |
|---|---|---|---|
| | ------Mg CO$_2$–C ha$^{-1}$ y$^{-1}$------- | | --- % --- |
| Root biomass regression | 6.0 (2.4) | 11.1 (1.0) | 54 (41) |
| Soil cores incubation | 0.4-1.9[c] | - | 8-17[d] |
| Hand-sorted root exclusion bags (HS) | 4.8 (0.3) | 6.1 (0.3) | 79 (3) |
| Intact root exclusion bags (IB) | 3.0 (0.3) | 6.1 (0.3) | 49 (7) |
| Soil $\delta^{13}$C-CO$_2$ natural abundance | - | - | 61 (39) |

Values are means and standard error, n = 22 for the root biomass regression, n = 47 for soil incubation,
n = 28 for both root exclusion bags techniques.
[a] Rh, heterotrophic respiration.
[b] Rs, total soil efflux taken alongside the Rh efflux.
[c] Efflux range at temperature between 14°C and 26°C.
[d] Calculated as Rh from incubation at 14°C and 26°C divided by average field Rs at 14°C and 26°C respectively.

Table 6: Qualitative evaluation of the partition methods

| Segregation method | Relative accuracy[a] | Precision[b] | Complexity of procedures[c] | Time needed to setup experiment[d] | Time needed to produce seasonal trends |
|---|---|---|---|---|---|
| Root biomass regression | High | Low | Simple | 2-3 days | 6 months to 1 year |
| Soil cores incubation | Low | High | Complex | 5-7 days | <1 to 2 months |
| Hand-sorted root exclusion bags (HS) | Intermediate | High | Simple | 2-3 days | 6 months to 1 year |
| Intact root exclusion bags (IB) | Intermediate | High | Simple | 2-3 days | 6 months to 1 year |
| Soil $\delta^{13}$C-$CO_2$ natural abundance | - | Low | Complex | 1-2 days | 6 months to 1 year |

[a] Partition methods that had Rh%: <10, 10-20 and >20 lower or larger than the $\delta^{13}$C-$CO_2$ natural abundance were categorized as high, intermediate and low accuracy, respectively.

[b] High, intermediate and low precision were attributed to Rh% standard errors of <10, 10-20 and >20, respectively.

[c] Methods with five steps or less were deemed simple and six steps or more deemed as more complex.

[d] The time needed to setup experiment was assessed with the number of working hours required prior to be able to start the measurements.

[e] The time needed to produce seasonal trends was the number of months of measurements required to characterize the Rh at the different temperature and moisture levels of the

545     year.

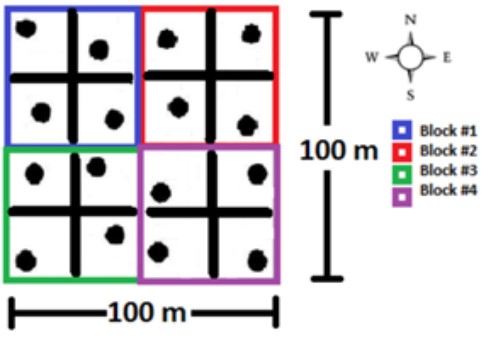

**Figure 1: Field sampling for the lab incubations, stratified random design in the 1 ha study area.**

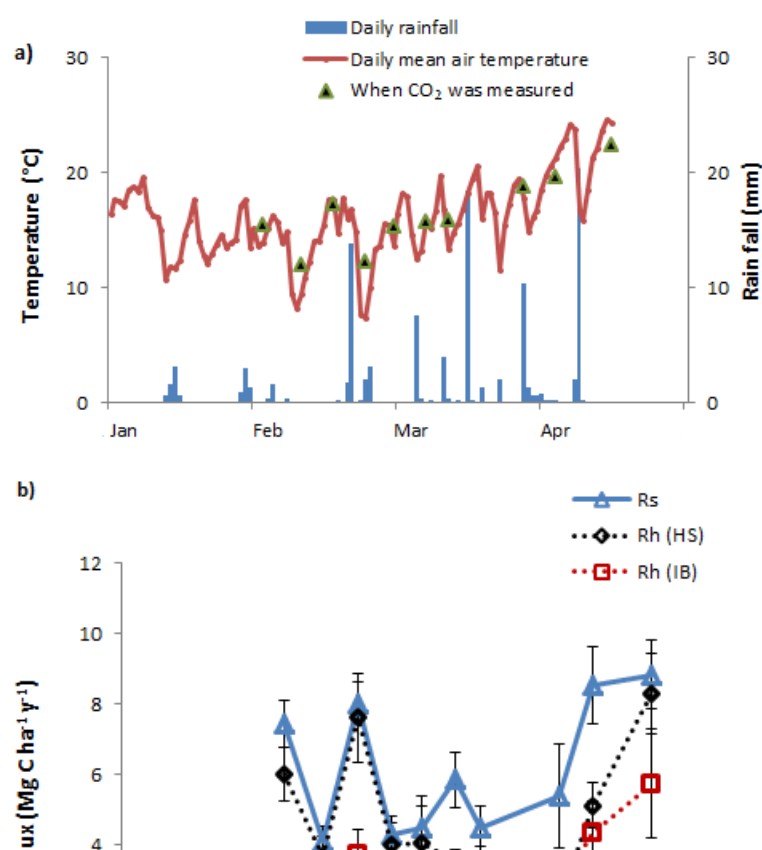

**Figure 2: a) Soil and air temperature and daily rainfall over the study period; b) Total soil $CO_2$ efflux (Rs), heterotrophic $CO_2$ efflux (Rh) from hand-sorted root exclusion bags (HS), and Rh from intact block root exclusion bags (IB).**

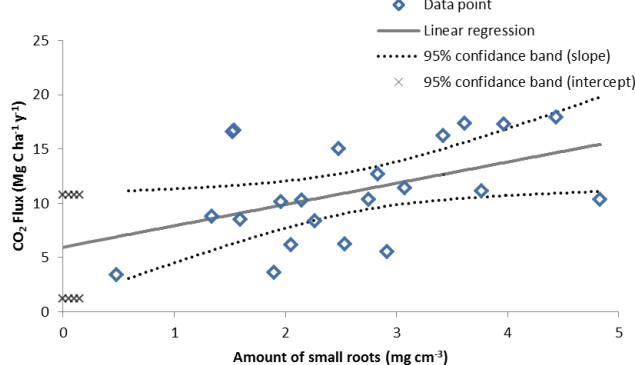

**Figure 3: Linear regression between root biomass and $CO_2$ efflux.**

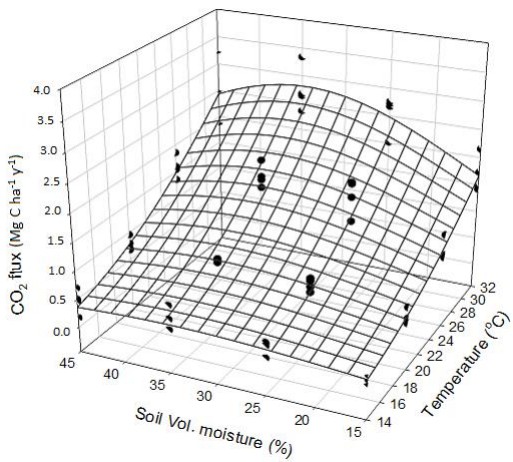

**Figure 4: Results from the lab incubation; regression between incubation temperature, moisture and $CO_2$ efflux.**

.