# Peer review of "Separation of soil respiration; a site-specific comparison of"

_SOIL, 2017_

## Short Comment (SC1) · 24 Jan 2018

In the Article "Separation of soil respiration; a site-specific comparison of partition methods" there is any mention to the partitioning of soil respiration using radiocarbon. Radiocarbon approach to estimate the heterotrophic efflux from soil represents an alternative and quite simple methodology. It is not invasive such other methodologies (e.g trenching) and with a single soil sampling campaign can allow to have an annual estimate of the heterotrophic efflux from soil. The methodology is quite well described in the following article:

Chiti T, Certini G, Forte C, Papale D, Valentini R (2016) Radiocarbon-Based Assessment of Heterotrophic Soil Respiration in Two Mediterranean Forests. Ecosystems,

19:62-72

A mention to the possibility of using this methodology should be reported in the text, were other approaches are reported.

The possibility of using radiocarbon it is also reported in the published reviews about soil respiration partitioning.

Please also note the supplement to this comment:
https://www.soil-discuss.net/soil-2017-38/soil-2017-38-SC1-supplement.pdf

[Figure]

**Supplement:**

[supplement omitted: unrelated document]

---

## Author Comment (AC1) · 27 Jan 2018

Thank you for the comment. The corrected version of the manuscript will discuss in the text that further comparative studies on soil $CO_2$ efflux segregation should also use the radiocarbon-based assessment (bomb-carbon) method as described by Chiti et al. (2015).
* * *

---

## Referee Comment (RC1) · Anonymous Referee #1 · 1 Feb 2018

It is not an easy task to separate autotrophic and heterotrophic soil respiration. Five methods were compared and none was superior. Accuracy was defined by the difference with the ÉÙ13C-CO2 natural abundance. This is a risky approach because this method has a low precision. Fortunately an second low precision method (root regression) gave a similar estimate of the ratio between heterotrophic and total soil respiration. Strengths and weaknesses of each method are discussed and it is concluded that a combination of methods is needed. This is a thorough investigation on an important subject. The manuscript is well written and easy to read. The methods section is relatively long, but this is inherent to a methodical paper. Non accessible PhD and BSc theses, such as Farmer 2013 and Tong 2015, can be removed from the literature references.

[Figure]

I found some typos: At line number 13 still many uncertainties and unknown. Should be unknowns. 53 Review have been made. Should be reviews. 120 to quantify and $CO_2$. Something missing? 172 AND 173 time inversion. What is an inversion of time? 223 potential air contaminations have to be considering. Considered. 239 could contains some portions of the roots respiration. Contain. 248 was statistically equal than. Equal to. 377 asindependent. Add space. 387 deciduousforest. Add space. 433 for 13 C Analysis. 13C.

---

## Referee Comment (RC2) · J. van Leeuwen (Referee) · 6 Feb 2018

The paper presents an interesting comparison of different respiration measures. I like the paper in general, and it's generally well written. I do however have a number of points on which the paper can be improved for better understanding and readability.

To begin with, please have a good look at the order of paragraphs in methods, results and discussion. Currently the used respiration methods are discussed in random order. I would suggest to follow a consistent order, for example following tables 6 and 7 in all three sections, so starting with regression, followed by lab incubation, root exclusion bags and finish with d13 C. This would improve the readability of the paper.

The five used respiration measures are mentioned in the abstract (lines 15-17), but the

introduction ends with simply stating that five methods were compared without further elaboration. Please explain the used methods here as well (for example in line 56), with the underlying thoughts on why these specific methods were selected.

At the start of the methods section, after describing the site selection, please start with soil characterization (lines 151-160). Also move the soil classification (line 177) to this section.

It could be helpful for understanding the sampling procedure followed in the lab incubation method if a sampling schema is added as figure, as currently it's a bit difficult to follow what happens with the various groups of samples (starting with 16 from the field, and in line 196 there are 22 samples?). In addition, if I read it correctly, the same samples went sequentially from a low incubation temperature to a high temperature. In the ideal scenario, the samples would be divided over the different incubation temperatures and studied independently. In the current setting dependency of the samples could be argued (repeated measurements from the same samples), but more importantly it adds uncertainty in terms of C present in the soils and potential shifts in the microbial community during the incubation (i.e. adapting to lower resource availability). Please include these limitations in the discussion of the used method.

Please remove Table 3, as it contains only information that is fully described in the text already (lines 197-198).

Also, please use consistent units for respiration throughout the paper, so either Mg C ha-1 y-1 or g m$^2$ h-1 to improve comparability of the various measures.

Some more specific comments: L124 (equation 1): please specify what f(x,y) represents (CO2 efflux)

L150: In the site selection you describe only forest vegetation. What do you mean with different landforms here?

L177: Please move this sentence to the method section on soil characterization.

L195-203: Please split this section according to the respiration measures that you discuss, as regression and incubation are presented as separate methods in tables and in the rest of the paper.

L199-200: please check carefully, I assume that you mean that Rh represents 54% of Rs, so I think the first time Rs in the sentence should be Rh.

L201-202: are these $CO_2$ efflux values for all moisture groups combined?

L208: The notably large .... respiration components. Please remove this sentence as it's trivial information.

L245: When I look at figure 1b I don't see an equilibrium in any of the groups. There seems to be the variation in respiration following the changes in temperature, but it does not show stability. Perhaps use the wording careful here.

L211-212: It is unclear where these values are coming from, as I can't find or derive them from table 6?

L323: I find "minimally disturbed microcosms" a bit doubtful here, as soils were sorted and repacked in different vials in the lab. Keeping the soil and the larger aggregates intact during such a procedure is hardly possible, I assume that you refer here to undisturbed microaggregates?

Minor textual comments:

L64: surveyed -> found

L102: report -> analysis

L105: soil cores with a volume of

L120: the boxes were opened to vent

L140: previous to -> before

L180: sub-superficial -> sub-surface or sub-soil

L184: based on previous results

L185: type of parent material and level of mineralization of the bedrock

L187:Split sentence in two, the first discussing climate, the second discussing respiration.

L252: rephrase for readability: Although not statistically significant, the maximum Rh in the HS and IB bags was found at a relatively low moisture content (9.5 and 21.4%, respectively, Table 4).

L262: please remove "In this study the regression had ten points (45%) outside the confidence interval but". The rest of the sentence is a repetition of L197-198, so either generalize for discussion or remove as well.

L295: variance -> variation

L298: in this regard

L300: to standardize

L338: variance of -> variation in

---

## Author Comment (AC2) · 7 Feb 2018

Thank you for comments to improve our manuscript. I appreciate the valuable suggestions from this review process. The co-authors and I are currently revising the manuscript and carefully addressing the suggestions. We believe the clarity and quality of the manuscript will be improved with the changes.

---

## Author Comment (AC4) · 7 Mar 2018

I appreciate the positive comments and suggestions from the reviewers. I already addressed most subjects that were pointed-out (attached document) and currently my co-authors are reviewing the corrected/improved manuscript. The full response to the review should be finished soon.

Please also note the supplement to this comment:
https://www.soil-discuss.net/soil-2017-38/soil-2017-38-AC4-supplement.pdf

[Figure]

**Supplement:**

[revised manuscript text omitted]

**3.3 Root regression**

**The 22 quadrats used for the root regression assessment yielded average Rs of 0.46 ±0.04 g $CO_2$ $m^2$ $h^{-1}$. The regression of the $CO_2$ efflux against root density produced a statistically significant slope correlation of 0.08 ±0.04 g $CO_2$ $m^2$ $h^{-1}$ per mg root $cm^{-3}$ (p=0.03), and set the intercept at 0.25 ±0.10 g $CO_2$ $m^2$ $h^{-1}$ (p=0.02) which represented the basal efflux in absence of root i.e. the Rh (Fig. 2 and Table 3). The Rh measured when the root regression technique was performed (October 2016) was 6.0 ±2.4 Mg C $ha^{-1}$ $y^{-1}$, equivalent to 54% of the Rs (Table 6).**

**3.3 1lab incubation**

~~The 22 quadrats used for the root regression assessment yielded average Rs of 0.46 ±0.04 g $CO_2$ $m^2$ $h^{-1}$. The regression of the $CO_2$ efflux against root density produced a statistically significant slope correlation of 0.08 ±0.04 g $CO_2$ $m^2$ $h^{-1}$ per mg root $cm^{-3}$ (p=0.03), and set the intercept at 0.25 ±0.10 g $CO_2$ $m^2$ $h^{-1}$ (p=0.02) which represented the basal efflux in absence of root i.e. the Rh (Fig. 2 and Table 3). The Rs measured when the root regression technique was performed (October 2016) was 11.1 ±1 Mg C $ha^{-1}$ $y^{-1}$ (Table 6), equivalent to 54% of the Rs.~~

During the incubation with minimally disturbed soil microcosms, the average (moisture levels combined) $CO_2$ efflux at 14, 20, 26 and 32ºC was 0.136 +0.50, 0.67 +0.38, 0.1.40 +0.91 and 2.24 +1.39 Mg $CO_2$m$h^{-1}$, respectively (Fig. 3). The exponential relationship between $CO_2$ efflux, soil temperature and moisture is presented in Table 4.

**3.4 Root exclusion bag methods**

During the root exclusion bags measurements period (Feb-Apr 2017), the average air temperature was 16ºC and the total rainfall 107 mm. During that period the Rs averaged 6.1 Mg C $ha^{-1}$ $y^{-1}$ (Fig. 1). One of the requirements for the suitability of root exclusion bag methods to estimate Rh is that soil bulk density, soil temperature and moisture are statistically equal inside and outside of the bags. In this experiment, no significant differences were detected regarding the bulk density and soil temperature (p=0.87 and p=0.15, respectively) but the volumetric soil moisture in the HS bags was on average 17% lower than outside the root exclusion bags (p=0.04) (Table 2). As would be expected, all Rh IB

and Rh HS efflux rates were lower than the Rs efflux at each measurement date. Throughout the experiment, the Rh IB was repetitively lower than the Rh HS except on March 31[st] (Fig. 1b).

3.3 Root regression and lab incubation

The 22 quadrats used for the root regression assessment yielded average Rs of $0.46 \pm 0.04$ g $CO_2$ m$^{-2}$ h$^{-1}$. The regression of the $CO_2$ efflux against root density produced a statistically significant slope correlation of $0.08 \pm 0.04$ g $CO_2$ m$^{-2}$ h$^{-1}$ per mg root cm$^{-3}$ (p = 0.03), and set the intercept at $0.25 \pm 0.10$ g $CO_2$ m$^{-2}$ h$^{-1}$ (p = 0.02) which represented the basal efflux in absence of root i.e. the Rh (Fig. 2 and Table 3). The Rs measured when the root regression technique was performed (October 2016) was $11.1 \pm 1$ Mg C ha$^{-1}$ y$^{-1}$ (Table 6), equivalent to 54% of the Rs.

During the incubation with minimally disturbed soil microcosms, the average $CO_2$ efflux at 14, 20, 26 and 32°C was $0.0151 \pm 0.021$, $0.0282 \pm 0.016$, $0.0585 \pm 0.038$ and $0.0938 \pm 0.058$ g $CO_2$ m$^{-2}$ h$^{-1}$, respectively (Fig. 3). The exponential relationship between $CO_2$ efflux, soil temperature and moisture is presented in Table 4.

**3.54 Soil $\delta^{13}$C-$CO_2$ natural abundance**

The $\delta^{13}$C-$CO_2$ natural abundance determination satisfactorily segregated the three respiration components (Table 5). The fact that the $\delta^{13}$C-$CO_2$ of the Rh HS, Rh IB and Rh lab were in a very close range indicated that in the field the efflux measured in the root exclusion bags were not contaminated with root respiration. Based on the $\delta^{13}$C-$CO_2$ of the Rs, the Rh lab and the Ra lab the percentage of heterotrophic respiration was $61 \pm 39\%$ (Table 6). The notably large standard error of the percentage of heterotrophic respiration was due to the large variance in the $\delta^{13}$C-$CO_2$ of the three respiration components.

Using the $\delta^{13}$C-$CO_2$ method as base-line, the percent increase/decrease of Comparing with the Rh from the $\delta^{13}$C-$CO_2$ method, the root 
[revised manuscript text omitted]
 3 and 6, respectively). Although not statistically significant, the HS and IB soil moisture parameter in the regression fit (i.e. y0, Table 4) showed that maximum Rh was when moisture content was relatively low (9.5 and 21.4%, respectively). Accordingly, this could partly explain the larger HS Rh efflux. In addition, the breakdown of numerous soil aggregates during the root removal likely allowed the soil microorganisms to thrive in previously encrusted SOM domains of the HS soil. It has been shown that the part of the SOM that is located in the interior of the soil aggregates is hardly accessible to microorganisms, and thus not easily mineralized unless the aggregates are shattered (Goebel et al., 2005).

**4.3 Root and carbon dioxide efflux regression technique**

As demonstrated by Gupta and Singh (1981) the intercept of the regression line between the independent variable (i.e. root biomass) and the dependent variable (i.e. Rs) corresponds to soil respiration in absence of root (i.e. Rh) (Fig. 2). In this study the regression had ten points (45%) outside the confidence interval but the intercept ($0.25 \pm 0.10$ g $CO_2$ m$^{-2}$ h$^{-1}$) and slope ($0.08 \pm 0.04$ g $CO_2$ X mg root cm$^3$) were still statistically significant (P = 0.02 and 0.03, respectively) (Fig. 2, Table 4). These large coefficients of variance caused the largest standard error value in the Rh/Rs assessment (54 $\pm 41$ %, Table 6). The uncertainty in the regression fit was likely caused in large part by the older roots which are bulkier but respire less than fine and young roots (Behera et al., 1990). However, this method had the closest average Rh/Rs to the $\delta^{13}C$ natural abundance technique. Consequently the root regression technique was assessed as high accuracy and low precision (Table 7). Previous studies also highlighted large variation of $CO_2$ efflux and root biomass which causes relatively low coefficient of determinations (Behera et al., 1990; Farmer, 2013). In accordance to Kuzyakov (2005), this method was comparatively simple (Table 7).

**4.4 Lab incubation method**

Interpreting soil respiration processes in response to seasonal changes is generally challenging because soil temperature and moisture regularly covary (Carbone et al., 2011; Davidson et al., 1998). The lab incubation technique was the only method capable of dividing the effect of soil temperature and moisture on Rh and to produce a significant Gaussian regression fit (Table 4). However, the microcosm incubation produced Rh values notably lower than the other techniques (Table 6). This might be due to the fact that the soil column in the incubation microcosms were 5 cm high while the A horizon in the field (i.e. where the Rh assessments from the other techniques were made) was 10 cm thick (Table 2). Further studies should test the effect of microcosm height on Rh in relation to field measurements. The low Rh of the lab incubation method could also be attributed in part to the fact that this technique did not contain any rhizomicrobial respiration and its priming effect. That is, this method produced Rh from basal microbial respiration which is considered to be from stabilized SOM with slow turnover rates (Kuzyakov, 2006 Neff et al., 2002). In view of that, with additional field and lab methods development it would be possible to further segregate Rh assessments into percentage of rhizomicrobial respiration, decomposition of plant residues and basal decomposition of SOM. Overall, the lab incubation technique was slightly more complex than the non-isotopic field Rh assessment methods but allowed a prompt determination of Rh whilst simulating year round field environment (Table 7).

**4.5 Comparison of methods and recommendations**

The analysis of the five different Rh/Rs partitioning methods examined in this study shows that none of them was fully satisfactory. That is, each technique had strengths and weaknesses (Table 76). Using $\delta^{13}$C-$CO_2$ is acknowledged as the preeminent way to segregate Rh/Rs (Cheng, 1996; Kuzyakov, 2006); and accordingly accuracy was defined by the difference with this method. We recognize this was a precarious approach because the $\delta^{13}$C-$CO_2$ method had a large variation. Fortunately, the root regression, which is also recognized in the literature as a reliable method (Kuzyakov, 2006), gave a similar %Rh estimate. However, we found several other shortcomings with the $\delta^{13}$C-$CO_2$ method. First, the conjunction of field and lab procedures makes it difficult to complete this method in one day as needed. Second, the air flushing with $CO_2$ free gas in the field (to prevent ambient $\delta^{13}CO_2$ contamination) makes that technique more complex than the other methods to assess Rh%. Third, the ability to perform this technique in remote areas is limited because the $\delta^{13}$C-CO$_2$ needs to be quickly assessed with a calibrated and accurate spectrometer (Midwood et al., 2006). Fourth, the large variation in $\delta^{13}$C-CO$_2$ of the respiration components (i.e. Ra, Rh and Rs)

impeded the assessment of Rh% per individual collar. Accordingly, further studies should analyze the spatial relationships of $\delta^{13}$C-CO$_2$ with soil properties and root characteristics. As standalone, the $\delta^{13}$C-CO$_2$ technique was unable to produce assessment of soil CO$_2$ efflux; thus needed to be performed in conjunction with field Rs measurements. In this regard, the $\delta^{13}$C-CO$_2$ complemented well with root exclusion bags methods because it allowed to determine if the buried bags had teared and been invaded by roots and to standardize Rh% determination.

The root regression method had the advantage to be simple, to produce an average Rh% close to the $\delta^{13}$C-CO$_2$ natural abundance and the disadvantage to require a high number of replicates due to low coefficient of determination between CO$_2$ efflux and root biomass. Another disadvantage of the root regression technique is that in order to produce seasonal trends, the labor intensive procedures (i.e. pit digging, CO$_2$ measurements and root counting) need to be reinitiated several times during the years. This shortcoming can be particularly impractical in small plot experiments.

Complementary studies should assess thresholds of root size to be included in the regression fit in order to optimize the correlation fit and use the $\delta^{13}$C-CO$_2$ natural abundance method to determine the effect of root size on the isotopic signature.

The root exclusion bags methods (i.e. HS and IB) had the advantage to be easy to monitor throughout the year. That is, because the % of Rh is unlikely to be constant in time it is important to assess it periodically. The bags methods can be considered as a miniaturization of the traditional soil trenching method. However, contrasting with large trenches (e.g. Comeau et al., 2016; Fisher and Gosz, 1986) the root exclusion bags had the advantage to be simpler to establish and to allow mycorrhiza development inside the mesh bags (Moyano et al., 2007). Conversely, due to the relatively small bag sizes, root webs on the outside edge could potentially contaminate Rh assessment. In this study, the $\delta^{13}$C-CO$_2$ determination made with the collars located in the center of the bags showed no isotopic signature of root respiration.

Similarly with the trenching method, the root exclusion bag methods had the disadvantages to require several months of soil stabilization before starting CO$_2$ efflux measurements. 
[revised manuscript text omitted]

.

---

## Author Response (AR1)

Topical Editor Decision: Revision by Peter de Ruiter

Comments to the Author:

*Dear Dr. Comeau:*

*I have seen the comments provided by the reviewers on your manuscript, and your response to these*
*comments. I understand that you are willing to address all comments in a new revised version of the manuscript. I kindly ask you to do so in order to accept the article for SOIL.*

*Kind regards, Peter de Ruiter*

**We would like to say thank you to the referees for the comments to improve our manuscript. We appreciate the valuable suggestions from this review process. The co-authors and I have revised the**
**manuscript and carefully addressed all the suggestions. We believe the clarity and quality of the manuscript has been enhanced with the changes.**

**Please find below our detailed response to all comments in bold font; the comments are in italics. Included in the revised submission is a document (below) with changes visible in Track Changes.**

Anonymous Referee #1

*It is not an easy task to separate autotrophic and heterotrophic soil respiration. Five methods were compared and none was superior. Accuracy was defined by the difference with the ÉU° 13C-CO2 natural abundance. This is a risky approach because this method has a low precision. Fortunately an second low precision method (root regression) gave a similar estimate of the ratio between heterotrophic and total*
*soil respiration. Strengths and weaknesses of each method are discussed and it is concluded that a combination of methods is needed. This is a thorough investigation on an important subject. The manuscript is well written and easy to read. The methods section is relatively long, but this is inherent to a methodical paper.*

**We appreciate the positive comments from the reviewer. To address the concern related to**
**"accuracy", the text and table 6 were modified to refer to "relative accuracy" because accuracy was measured in relation to the $\delta^{13}C\text{-}CO_2$ natural abundance method.**

*Non accessible PhD and BSc theses, such as Farmer 2013 and Tong 2015, can be removed from the literature references.*

**Regarding citations from Farmer (2013) and Tong (2015), the latter was removed from the literature**
**references as suggested and the former was modified in the reference section to allow reader to retrieve it. "Farmer, J.A.: Measuring and modeling soil carbon and carbon dioxide emissions from Indonesian peatlands under land-use change, Ph D, University of Aberdeen, UK, 2013. Available from University of Aberdeen library collections,**

https://aulib.abdn.ac.uk/F?func=direct&local_base=ABN01&doc_number=001582866" see lines 395-397.

*I found some typos:*

*At line number*

*13 still many uncertainties and unknown. Should be unknowns.*

**Addressed: "… still many uncertainties and unknowns …" see line 13.**

*53 Review have been made. Should be reviews.*

**Addressed: "Reviews of Rh-Rs segregation …" see line 53.**

*120 to quantify and CO2. Something missing?*

**Addressed: "The GC system was equipped with a flame ionization detector and an electron capture detector to quantify $CO_2$ concentration." see lines 118-119.**

*172 AND 173 time inversion. What is an inversion of time?*

**Addressed: "The time needed to set up the experiment was assessed by counting the number of working hours (eight hours equal one day) required prior to the start of the measurements." See lines 186-187.**

*223 potential air contaminations have to be considering. Considered.*

**Addressed: "… potential air contaminations have to be considered …" see lines 299-300.**

*239 could contains some portions of the roots respiration. Contain.*

**Addressed: " … this technique did not contain …" see lines 252-253.**

*248 was statistically equal than. Equal to.*

**Addressed: "…was statistically equal to …" see lines 278-279.**

*377 asindependent. Add space.*

**Addressed: "…temperature as independent …" see lines 408.**

*387 deciduousforest. Add space.*

**Addressed: "… deciduous forest …" line 419.**

*433 for 13 C Analysis. 13C.*

**Addressed: "… for 13C …" line 467.**

J. van Leeuwen (Referee)

jeroen.vanleeuwen@wur.nl

*The paper presents an interesting comparison of different respiration measures. I like the paper in*
*general, and it's generally well written. I do however have a number of points on which the paper can be improved for better understanding and readability.*

*To begin with, please have a good look at the order of paragraphs in methods, results and discussion. Currently the used respiration methods are discussed in random order. I would suggest to follow a consistent order, for example following tables 6 and 7 in all three sections, so starting with regression,*
*followed by lab incubation, root exclusion bags and finish with d13 C. This would improve the readability of the paper.*

**Addressed: we have corrected this inconsistence. The different sections, paragraphs and sentences related to the five methods used were rearranged to follow a consistent order. Specifically, the following order was standardized throughout the text and tables: root regression, soil cores**
**incubation, hand-sorted root exclusion bags (HS), intact root exclusion bags (IB) soil $\delta13C-CO_2$ natural abundance.**

*The five used respiration measures are mentioned in the abstract (lines 15-17), but the introduction ends with simply stating that five methods were compared without further elaboration. Please explain the used methods here as well (for example in line 56), with the underlying thoughts on why these specific*
*methods were selected.*

**Addressed: More detailed information about the specific methods selected were added at the end of the introduction; "The goal of our study was to compare partitioning methods to separate $CO_2$ efflux into its Rs and Rh component in a subtropical secondary forest in Hong Kong. Five methods were selected based on their suitability in the studied ecosystem. Three methods were traditional**
**techniques (i.e. regression between root mass and root derived $CO_2$; root exclusion bags with hand-sorted roots; and soil $\delta13C-CO_2$ natural abundance) and two were innovative variation of existing methods (i.e. root exclusion bags with intact soil blocks; and lab incubations with minimally disturbed soil microcosm cores). The influence of soil moisture and temperature on $CO_2$ efflux was also analyzed." See lines 54-60.**

*At the start of the methods section, after describing the site selection, please start with soil characterization (lines 151-160).*

**Addressed: section 2.1 is now "Soil general characterization" see line 73.**

*Also move the soil classification (line 177) to this section.*

**We recognize that stating the soil type name in the method section would beforehand help the**
**readers to understand the ecosystem were the study was performed. However, we consider that the**
**soil classification we did with the World Reference Base (WRB) system an integral part of the results**
**because it had not been done before in this area; and accordingly, further investigations in this region**
**will refer to our determination/classification.**

*It could be helpful for understanding the sampling procedure followed in the lab incubation method if a*
*sampling schema is added as figure, as currently it's a bit difficult to follow what happens with the*
*various groups of samples (starting with 16 from the field, and in line 196 there are 22 samples?).*

**Addressed: a figure showing the sampling procedure followed was added as new Fig 1. Also,**
**clarification was made in the text "… four groups of four soil cores (stratified random design) were**
**collected …" Lines 105-106.**

[Figure]

**Fig.1. Field sampling for the lab incubations, stratified random design in the 1 ha study area.**

*In addition, if I read it correctly, the same samples went sequentially from a low incubation temperature*
*to a high temperature. In the ideal scenario, the samples would be divided over the different incubation*
*temperatures and studied independently. In the current setting dependency of the samples could be*
*argued (repeated measurements from the same samples), but more importantly it adds uncertainty in*
*terms of C present in the soils and potential shifts in the microbial community during the incubation (i.e.*
*adapting to lower resource availability). Please include these limitations in the discussion of the used*
*method.*

**Addressed: The section 4.2 was extended and improved to clarify this potential issue; "to prevent**
**potential shifts in the microbial community during the incubations (i.e. adapting to lower resource**
**availability), prior to the beginning of the experiment the microcosms were left stabilize for two**
**weeks. Accordingly, the fresh and labile organic residues that would in the other segregation methods**
**contribute to the soil respiration had already decomposed before the beginning of the incubations."**
**lines 248-250.**

*Please remove Table 3, as it contains only information that is fully described in the text already (lines 197-198).*

**Addressed: Table 3 was removed.**

*Also, please use consistent units for respiration throughout the paper, so either Mg C ha-1 y-1 or g m2 h-1 to improve comparability of the various measures.*

**Addressed: To be consistent with gas flux units, now everything is presented as Mg C ha$^{-1}$ y$^{-1}$.**

Some more specific comments:

*L124 (equation 1): please specify what f(x,y) represents (CO2 efflux).*

**Addressed: "where *f(x,y)* is the CO$_2$ efflux function ..." see line 125.**

*L150: In the site selection you describe only forest vegetation. What do you mean with different landforms here?*

**Addressed: "... different landforms (i.e. back-slope and foot-slope) present at the site ..." see line 74.**

*L177: Please move this sentence to the method section on soil characterization.*

**See response to "*Also move the soil classification (line 177) to this section.*" comment above.**

*L195-203: Please split this section according to the respiration measures that you discuss, as regression and incubation are presented as separate methods in tables and in the rest of the paper.*

**Addressed: this section was divided into 3.2 and 3.3, see lines 201-208.**

*L199-200: please check carefully, I assume that you mean that Rh represents 54% of Rs, so I think the first time Rs in the sentence should be Rh.*

**Addressed: This sentence was corrected "The Rh measured when the root regression technique was performed (October 2016) was 6.0 ±2.4 Mg C ha-1 y-1, equivalent to 54% of the Rs (Table 5)." Lines 206-207.**

*L201-202: are these CO2 efflux values for all moisture groups combined?*

**Addressed: " ... the average (moisture levels combined) CO$_2$ efflux ..." see line 209.**

*L208: The notably large .... respiration components. Please remove this sentence as it's trivial information.*

**Addressed: This sentence was removed.**

*L245: When I look at figure 1b I don't see an equilibrium in any of the groups. There seems to be the variation in respiration following the changes in temperature, but it does not show stability. Perhaps use the wording careful here.*

**Addressed: section 4.3 was amended and improved to clarify what we meant with the term "equilibrium"; "After the three months of soil stabilization period, both bag methods for partitioning total soil respiration and root-free soil respiration components successfully produced Rs>Rh in every sampling dates indicating that efflux rates within the bags had reached an apparent post disturbance state (Fig. 1)." lines 273-276.**

*L211-212: It is unclear where these values are coming from, as I can't find or derive them from table 6?*

**Addressed: This sentence was re-written to improve clarity "Using the $\delta$13C-CO2 method as base-line, the percent increase/decrease of the root regression, lab incubation, hand-sorted and intact block (IB) root exclusion techniques were -11%, -72-87%, +30% above and -20%, respectively (Table 5)." Lines 227-229.**

*L323: I find "minimally disturbed microcosms" a bit doubtful here, as soils were sorted and repacked in different vials in the lab. Keeping the soil and the larger aggregates intact during such a procedure is hardly possible, I assume that you refer here to undisturbed microaggregates?*

**Addressed: section 2.3 was amended to specify it is the micro-aggregate that should not be disturbed/broken. "special care to not destroy the micro-aggregates" Lines 107-108.**

Minor textual comments:

*L64: surveyed -> found*

**Addressed: "the following genera were found …" see lines 67-68.**

*L102: report -> analysis*

**Addressed: "A linear regression analysis …" see line 100**

*L105: soil cores with a volume of*

**Addressed: "undisturbed soil cores with a volume of …" see line 104.**

*L120: the boxes were opened to vent*

**Addressed: "each measurement session, the boxes were opened to vent …" see line 120.**

*L140: previous to -> before*

**Addressed: " … 90 minutes before …" see line 164.**

*L180: sub-superficial -> sub-surface or sub-soil*

**Addressed: "The sub-surface soil …" see line 185.**

*L184: based on previous results*

**Addressed: "…status based on previous results …" see line 195.**

*L185: type of parent material and level of mineralization of the bedrock*

**Addressed: " … type parent material and level of mineralization of the bedrock …" see lines 199-200.**

*L187: Split sentence in two, the first discussing climate, the second discussing respiration.*

**Addressed: "During the root exclusion bags measurements period (Feb-Apr 2017), the average air temperature was 16ºC and the total rainfall 107 mm. During that period the Rs averaged 6.1 Mg C ha-1 y-1 (Fig. 1)" see lines 213-214.**

*L252: rephrase for readability: Although not statistically significant, the maximum Rh in the HS and IB bags was found at a relatively low moisture content (9.5 and 21.4%, respectively, Table 4).*

**Addressed: "… supported by the fact that in the regression fit the maximum Rh was when moisture content was relatively low …" see lines 283-284.**

*L262: please remove "In this study the regression had ten points (45%) outside the confidence interval but". The rest of the sentence is a repetition of L197-198, so either generalize for discussion or remove as well.*

**Addressed: section 3.2 was shortened to remove the redundancies.**

*L295: variance -> variation*

**Addressed: "Fourth, the large variation …" see line 319.**

*L298: in this regard*

**Addressed: "In this regard, the $\delta^{13}$C-CO$_2$ …" see line 323.**

*L300: to standardize*

**Addressed: " … by roots and to standardize …" see line 324.**

*L338: variance of -> variation in*

**Addressed: "…results showed large variation in …" see lines 364-365.**

[revised manuscript text omitted]
 3 and 6, respectively). Although not statistically significant, the HS and IB soil moisture parameter in the regression fit (i.e. y0, Table 4) showed that maximum Rh was when moisture content was relatively low (9.5 and 21.4%, respectively). Accordingly, this could partly explain the larger HS Rh efflux. In addition, the breakdown of numerous soil~~

aggregates during the root removal likely allowed the soil microorganisms to thrive in previously encrusted SOM domains of the HS soil. It has been shown that the part of the SOM that is located in the interior of the soil aggregates is hardly accessible to microorganisms, and thus not easily mineralized unless the aggregates are shattered (Goebel et al., 2005).

**4.3 Root and carbon dioxide efflux regression technique**

As demonstrated by Gupta and Singh (1981) the intercept of the regression line between the independent variable (i.e. root biomass) and the dependent variable (i.e. Rs) corresponds to soil respiration in absence of root (i.e. Rh) (Fig. 2). In this study the regression had ten points (45%) outside the confidence interval but the intercept ($0.25 \pm 0.10$ g $CO_2$ m$^2$ h$^{-1}$) and slope ($0.08 \pm 0.04$ g $CO_2$ X mg root cm$^3$) were still statistically significant (P = 0.02 and 0.03, respectively) (Fig. 2, Table 4). These large coefficients of variance caused the largest standard error value in the Rh/Rs assessment ($54 \pm 41$ %, Table 6). The 
[revised manuscript text omitted]

.